# Lessons about Botulinum Toxin A Therapy from Cervical Dystonia Patients Drawing the Course of Disease: A Pilot Study

**DOI:** 10.3390/toxins15070431

**Published:** 2023-06-30

**Authors:** Harald Hefter, Isabelle Schomaecker, Max Schomaecker, Beyza Ürer, Raphaela Brauns, Dietmar Rosenthal, Philipp Albrecht, Sara Samadzadeh

**Affiliations:** 1Department of Neurology, University of Düsseldorf, Moorenstrasse 5, 40225 Düsseldorf, Germanyphil.albrecht@gmail.com (P.A.); sara.samadzadeh@yahoo.com (S.S.); 2Department of Neurology, Maria Hilf Clinics, 41063 Moenchengladbach, Germany; 3Charité—Universitätsmedizin Berlin, Corporate Member of Freie Universität Berlin and Humboldt-Universität zu Berlin, Experimental and Clinical Research Center, 13125 Berlin, Germany; 4Department of Regional Health Research and Molecular Medicine, University of Southern Denmark, 5230 Odense, Denmark; 5Department of Neurology, Slagelse Hospital, 4200 Slagelse, Denmark

**Keywords:** cervical dystonia, course of disease (CoD), CoD graphs, botulinum toxin therapy, long-term treatment, secondary treatment failure, primary treatment failure

## Abstract

Aim of the study: To compare the course of severity of cervical dystonia (CD) before and after long-term botulinum toxin (BoNT) therapy to detect indicators for a good or poor clinical outcome. Patients and Methods: A total of 74 outpatients with idiopathic CD who were continuously treated with BoNT and who had received at least three injections were consecutively recruited. Patients had to draw the course of severity of CD from the onset of symptoms until the onset of BoNT therapy (CoDB graph), and from the onset of BoNT therapy until the day of recruitment (CoDA graph) when they received their last BoNT injection. Mean duration of treatment was 9.6 years. Three main types of CoDB and four main types of CoDA graphs could be distinguished. The demographic and treatment-related data of the patients were extracted from the patients’ charts. Results: The best outcome was observed in those patients who had experienced a clear, rapid response in the beginning. These patients had been treated with the lowest doses and with a low number of BoNT preparation switches. The worst outcome was observed in those 17 patients who had drawn a good initial improvement, followed by a secondary worsening. These secondary nonresponders had been treated with the highest initial and actual doses and with frequent BoNT preparation switches. A total of 12 patients were primary nonresponders and did not experience any improvement at all. No relation between the CoDB and CoDA graphs could be detected. Primary and secondary nonresponses were observed for all three CoDB types. The use of initial high doses as a relevant risk factor for the later development of a secondary nonresponse was confirmed. Conclusions: Patients’ drawings of their course of disease severity helps to easily detect “difficult to treat” primary and secondary nonresponders to BoNT on the one hand, but also to detect “golden responders” on the other hand.

## 1. Introduction

Idiopathic focal cervical dystonia (CD) is a chronic neurological disorder that typically manifests clinically between the ages of 40 and 60 years and affects women 1.5–1.9 times more often than men [1,2]. The clinical spectrum of symptoms ranges from mild muscular tension in circumscribed head or neck muscle groups to abnormal head posture, severe involuntary jerks of head and neck muscles, and impaired head control [3], which can interfere with daily activities [4,5]. Repeated intramuscular injections of botulinum neurotoxin type A (BoNT-A) have become the treatment of choice for CD [6,7]. Currently, less than 20% of mildly affected CD patients remain without BoNT treatment for an extended period of time [8] and are treated with muscle-relaxing and pain medication and physiotherapy.

Treating CD with BoNT-A or botulinum neurotoxin type B (BoNT-B) involves a complex interaction between the patient and the treating physician [9]. The initiation of BoNT therapy does not follow a standardized procedure. The de novo patient presents with a variety of symptoms, and the physician responds with the choice of the BoNT preparation, total dose, and an injection scheme for how to distribute the total dose to different muscles. When the patient is seen for a follow-up visit, the use of standardized scales to assess improvement, such as the TSUI score [10] or the Toronto Western Spasmodic Torticollis Rating Scale (TWSTRS) [11], is helpful in adjusting the choice of preparation, total dose, and injection scheme. In our centre, physicians use the TSUI score for the assessment of the therapy and to try to improve the BoNT therapy until the patient has a TSUI score lower than 6.

Major reasons why effective and safe BoNT treatment is discontinued in up to 46% of CD patients include a too-low initial dose [12] and the injection of incorrect muscles [12,13,14]. We have introduced the CAP/COL concept and demonstrated that, in some patients, the additional injection of the deep neck muscles [15] will lead to additional improvement. Unfortunately, none of the current scales for CD specifically address the action of the deep neck muscles [16,17,18].

Patients with an initially low severity of CD require doses as high as those with moderate or high severity [12]. Otherwise, these patients do not experience enough benefit and tend to discontinue BoNT therapy after a few treatment cycles [12]. Starting with a recommended or standardized dose may be helpful for inexperienced injectors [19]. However, adjusting treatment parameters is absolutely necessary to optimize outcomes [9,12].

Because of the complexity of the (most often unconscious) interaction between CD patients and treating physicians during BoNT therapy, it is important to integrate and take into account the patient’s experience of the course of the disease (CoD) [9,19]. Our CD patients have therefore been trained to document the course of disease severity in a manner similar to that of patients with pain syndromes. The treating physician usually sees only a small segment of CD aspects that are relevant to the patient’s quality of life [4]. In a study on the quality of life in long-term-treated CD patients, a high correlation between the TSUI score and the stigmatization item on the CDQ24 questionnaire [20] was found, but not between the TSUI score and emotional well-being or social life aspects. (For details, see [5]).

A study showed that patients experienced a better quality of life when assessed with the CDQ24 at week 12 of BoNT therapy, as compared to week 4, when the peak effect of the first injection of 500 U aboBoNT-A has already declined [19]. This suggests that patients require a positive perspective for their quality of life. At week 12, patients had experienced the first injection and expected another one, while at week 4, they were uncertain about the progression of the disease [19]. Thus, it is crucial to discuss the patient’s experience of the disease severity and treatment perspectives during BoNT therapy [9,19].

A recent publication reported that patients were asked to draw a graph illustrating the development of CD severity from the onset of symptoms until the start of BoNT therapy (CoDB graphs), which revealed three main types of graphs [9]. Different types of CoDB graphs were associated with different times to therapy, different initial doses, different adjustments of dose during BoNT therapy, and different assessments of the outcome after long-term BoNT therapy [9]. This underscores the importance of the course of disease before BoNT therapy for later outcomes, similar to the impact of the time to therapy on long-term outcomes in deep brain stimulation (DBS) in primary dystonia [21,22].

In the present study, patients were asked to draw the course of disease severity after the start of BoNT therapy until recruitment into the study (CoDA graphs). The study aimed to determine whether different types of CoDA graphs could be distinguished and whether patients with different types of CoDA graphs are treated differently, assess the effect of treatment differently, and have different long-term outcomes.

## 2. Results

### 2.1. Demographic and Treatment-Related Data and Outcome Measures of All Patients

Data on the entire cohort (n = 74) are presented in Table 1 in the column “ALL”. The mean age of onset was 45.3 years; the female/male ratio was 1.9, and therefore typical for idiopathic cervical dystonia [1]. The mean time to therapy was 5.7 years, and the mean duration of treatment was 9.6 years. The mean initial TSUI score was 8.9, and the mean actual TSUI score at recruitment was 4.4, corresponding to an improvement of 51% in the mean. The dose was increased from 166 to 218 uDU. The mean improvement since onset of BoNT-A therapy assessed by the patients via a questionnaire (IMPQ) was 43%, and the mean improvement according to the CoDA graphs (IMPD) was 45.9%.

### 2.2. The Different Types of CoDA Graphs (RR, CR, PR, STF, and OR)

To compare the different graphs of the patients, the x-axis (= time axis) was normalized: time, t, was transformed into 10*t/DURS for the CoDB graphs, where DURS is the time to therapy. For the CoDB graphs, time, t, was transformed into 10*t/DURT where DURT is the time of therapy.

Compared to the drawing of the CoDB graphs, the drawing of the CoDA graphs appeared to be more difficult for the patients. Only 66 CoDA graphs could be used for further analysis (Table 2). The rapid response (RR) subgroup comprised 15 patients, the continuous response (CR) subgroup comprised 21 patients, the poor response (PR) subgroup comprised 12 patients, the secondary treatment failure (STF) group comprised 17 patients, and the other response (OR) subgroup included only 1 patient. (For details on these subgroups, see the Section 5). In Figure 1 (right side), all of the CoDA graphs from the three (RR, CR, and PR) subgroups are presented. In Figure 1 (left side), the average CoDA graph plus/minus the 1 standard deviation range is presented for these three subgroups. In Figure 2 (right side), all of the CoDA graphs of the STFI subgroup (upper part) and the STFII subgroup (lower part) are presented. (For details on these subgroups, see the Section 5). In Figure 2 (left side), the average CoDA graphs plus/minus the 1 standard deviation range is presented for the STFI subgroup and the STFII subgroup. In Figure 3 (right side), the only OR CoDA graph is shown. This graph shows an excellent response, but in two intermittent instances, a severe worsening of unknown origin occurred. This patient drew a CoDB graph showing the two distinct phases of worsening (Figure 3, left side). 

### 2.3. Differences in Outcome between Four Different Patient Subgroups

Patients were split-up into 4 subgroups (RR, CR, PR, and STF). The four-group ANOVA revealed significant (*p* < 0.009) differences for the two outcome measurements, the IMPQ and IMPD. (For details, see Table 1). 

In the RR group, the initial TSUI score was the highest (mean: 11.0), the actual TSUI score was the lowest (mean: 3.0), and thus, the IMPTSUI was the highest (mean: 7.9 = 72%), the initial dose was the lowest (mean: 151 uDU), the actual dose was the lowest (mean: 167 uDU), and the increase in the dose was the lowest (mean: 13.4 uDU). The mean IMPQ was 59%, and the mean IMPD was 61%. The majority of these patients were females (13 out of 15 = 87%). This group represents the best responders. In this group, only 4 out of 15 patients (<27%) had been switched to another BoNT-A preparation during a treatment duration of greater than 10 years for personal reasons, not because of the development of a clear-cut secondary treatment failure.

The time to therapy (mean: 48.5 months) was the lowest in the CR group; the mean IMPQ (61%) and the mean IMPD (64%) were the highest; the ATSUI was low (mean: 4.0), and the IMPTSUI was the second-best (mean: 4.7 = 55%). However, the dose (INDOSE) was increased from 161 to 233 uDU. A total of 7 out of 21 patients (<34%) were switched to another BoNT-A preparation for personal reasons or for cost-of-treatment aspects.

In the PR group, the time to therapy was the longest (mean DURS: 93 months (>7.7 years)) with a large variability (see Table 1). The mean initial severity of CD was the lowest (mean ITSUI: 6.0), and the rating of improvement by the treating physicians (IMPTSUI: 1.6 = 26.7%) and by the patients (IMPQ: 11.3%; IMPD = 7%) were the lowest. Half of these patients (6 out of 12) were switched to another BoNT-A preparation because of their poor response. 

The STF subgroup will be analysed in more detail below.

### 2.4. Special Analysis of the STF Subgroup and Switching of the BoNT-A Preparation

Patients in the STF group were the youngest (mean AGE: 57.8 years), had the lowest age at onset of symptoms (mean AOS: 43.0 years), were treated with the highest initial doses (mean IDOSE: 185 uDU), had the highest increase of dose (mean INDOSE: 72 uDU), and had the highest actual dose (mean ADOSE: 249 uDU), but they had the worst long-term outcome (mean ATSUI: 5.5). Improvement parameters (IMPTSUI = 3.9 = 39%; IMPQ = 33%; IMPD = 35%) were low, but better than in the PR group (see Table 1). This is the only subgroup that contained more males than females (9/8), and 9 of the 17 patients (>52%) had been switched to another BoNT-A preparation because of secondary worsening.

The STF group’s patients could be divided into subgroup I (STFI; n = 11), who did not show any further improvement after the secondary worsening had started and the BoNT-A preparation had been switched (Figure 2; upper part; IMPD = 25.4%), and subgroup II (n = 6), who responded with a clear-cut second improvement to the switching of the BoNT-A preparation (Figure 2; lower part; IMPD = 51.9%). A total of 4 of these 6 patients had been switched to incoBoNT-A, one from abo- to onaBoNT-A, and one from ona- to aboBoNT-A. 

A total of 10 of the 17 patients of the STF group (= 58.8%) had also been included in another study on neutralizing antibody (NAb) formation. All of these 10 STF patients had a positive mouse hemidiaphragm test (MHDA). The MHDA is a highly sensitive laboratory test used to detect neutralizing antibodies in the serum of BoNT-treated patients. The sera of our patients were analysed in the Toxogen^®^ laboratory (MMH, Hannover, Germany).

### 2.5. Comparison of CoDB and CoDA Graphs

When each of the three (RO, CO, and DO) CoDB graph subgroups was further subdivided into four different subgroups according to the type of CoDA graph (Table 2), Friedman´s test did not detect a significant relation between a special CoDB and a special CoDA graph. (For details on these subgroups, see the Section 5). The percentage of patients with an STF CoDA graph was the highest in the RO group (5/14 = 36%) and the lowest in the DO group (6/29 = 21%).

## 3. Discussion

### 3.1. Lesson 1: The “Staircase-like” Improvement of CD with Repeated BoNT Injections Every 3 Months

The general background and basis for the present results is the special plan for injection management at our institution, which keeps the duration of the injection cycle constant, with reinjections of the patients every 12 to 13 weeks [12]. This implies that most patients are reinjected well before the clinical effect of the previous injection has fully declined, and that patients experience a “staircase-like” improvement, injection by injection. This phenomenon is presented in detail in [23]. The outcomes of the present study do not include the analysis of the development of the peak effect at approximately four weeks after injection, but rather a continuous, stable plateau of improvement which is achieved via repeated injections at a fixed interval of 3 months and an analysis of the severity of CD at the end of an injection cycle.

### 3.2. Lesson 2: Most CD-Patients Respond Well to Repeated BoNT Injections Every 3 Months

Patients´ drawings of the course of CD severity after the onset of botulinum toxin therapy (CoDA graphs) differ considerably. A total of 54 out of 66 patients (=81.8%; RR + CR + STF + OR subgroups) experienced a good response during the initial phase of treatment. This corresponds with previously reported initial responder rates of more than 70%, with an improvement of more than 25% [19,24]. In a review from 2003, a benefit from BoNT therapy was reported in 70–85% of CD patients [25]. The long-term outcome from BoNT therapy depends on a variety of factors: the genetic constellation of the patient (see next paragraph), the BoNT preparation used [23,26], the dose per session, the length of the treatment cycle, the clinician’s knowledge about optimal injection sites for muscles (detailed in [27,28]), and the duration of the treatment [26]. 

### 3.3. Lesson 3: There Are “Golden Responders”, but the Patient´s and Physician’s Assessments May Be Different

Patients in the RR group (15 out of 66 = 23%) had the highest mean initial score (mean ITSUI = 11.0), the lowest remaining severity (mean ATSUI = 3.1), by far the best improvement in TSUI score (mean IMPTSUI = 7.9), were treated with the lowest initial dose (mean IDOSE = 151 uDU), had the lowest actual doses (mean ADOSE = 167 uDU), and needed the lowest increase of dose (mean INDOSE = 13 uDU; see Table 1). Most of these “golden responders” were females (87%). However, patient assessments were not the best in the RR group. Patients in the CR group (21 out of 66 = 32%) had the highest mean assessment of improvement (mean IMPQ = 60.7%; IMPD = 64%). These patients were satisfied with BoNT therapy and thus had the longest mean duration of therapy (130 months = 10.8 years) [14,29].

Similar to the finding that some special HLA classes are at a greatly increased risk of cervical dystonia, as in Caucasian Americans [30], or special forms of dystonia [31,32], it has been suspected that patients that are highly sensitive to BoNT therapy, who have been described previously [23], might have special MHC class constellations [23,33,34]. However, this has not been clearly demonstrated so far. 

### 3.4. Lesson 4: There Are Primary Nonresponders

Only 12 out of 66 patients (18%) drew a poor long-term outcome (PR subgroup) and were classified as primary nonresponders. Patients in the PR group had the lowest initial severity of CD (mean ITSUI = 6.0), the least improvement in TSUI score (mean IMPTSUI = 1.6) and were assessed as experiencing the least improvement (mean IMPQ = 11.3%, mean IMPD = 7%). They had the longest time to therapy (mean DURS = 93 months = 7.75 years) and the shortest duration of treatment (mean DURT = 75 months = 6.25 years). The obvious reason is that adherence to therapy is low when the efficacy of BoNT treatment is low [14,29]. 

The negative impact of a low initial severity of CD on later outcomes has recently been described for a large cohort of more than 300 BoNT naïve patients [12]. The results in the present PR subgroup are in full agreement with this observation. 

### 3.5. Lesson 5: Secondary Treatment Failure (STF) and NAb Formation in STF Is Frequent

During long-term treatment of more than 118 months (=9.9 years) in the mean, 17 patients (=25.8%) experienced a secondary treatment failure (STF), resulting in an incidence of STF of about 2.6%/year. In 5 of these patients, the bSTF was complete (cSTF: 5/66 = 7.6%), which is slightly more than the 5.9% reported by Walter et al. [35]

A total of 10 of these 17 patients who had an STF graph were selected for MHDA testing. All ten patients were MHDA-positive. Thus, the percentage of MHDA-positive patients among these 17 patients with STF was at least 58.8%. This finding is consistent with the observation that at least half of the patients with a suspected secondary treatment failure are MHDA-positive [36,37]. However, it must be kept in mind that systematic cross-sectional testing would have yielded higher percentages of MHDA-positive patients, as in other cross-sectional studies [26].

### 3.6. Lesson 6: Switching to Another BoNT-A Is a Therapy Option in Patients with STF

After an initial response as good as that in patients in the RR and CR groups, STF patients experienced a secondary worsening. A total of 9 of the 17 patients were switched to another BoNT-A preparation; six of them (>66%; STFII subgroup) responded to this switch very well. The efficacy of BoNT therapy after switching to another BoNT-A preparation in the STF subgroup was about 40% (mean IMPTSUI: 3.9; =39.8% of ITSUI) and was rated to be about 33% by the patients. (For details, see Table 1). In an even larger group of 64 CD patients with STF (after previous treatment with the complex-protein-containing abo- or onaBoNT-A preparation), the switch to incoBoNT-A (which is free of complex proteins and inactive, but immunologically relevant, fragments of BoNT-A) led to a similar improvement of about 30%, which was even more pronounced in the MHDA-negative patients [38]. 

In patients with STF (after therapy with a complex-protein-containing BoNT-A preparation), the switch to another BoNT preparation, such as BoNT-B [39,40] or BoNT-F [41,42] was tried years ago, but without satisfactory long-term effects. Therefore, cessation of BoNT-A therapy has been recommended, either forever or until NAb titres have fully declined [43,44], or a DBS operation is considered [22]. However, evidence has been presented that, in patients with STF (after abo- or onaBoNT-A treatment), the continuation of BoNT-A therapy with the complex-protein-free incoBoNT-A preparation Xeomin^®^ leads to a long-lasting improvement [38] and a decline in NAbs [45]. Whether a DBS operation is superior to switching from abo- or onaBoNT-A to incoBoNT-A in patients with STF remains to be demonstrated.

### 3.7. Lesson 7: The Use of Initial High Doses Is a Risk Factor for the Development of STF and NAb Induction

Patients in the STF group were the youngest patients in the study and had the lowest age at manifestation of CD. They were treated with the highest initial doses, had the highest increase of dose, and the worst outcome (see Table 1). This is in line with the hypothesis that NAbs may be induced early [46]. There is some evidence that patients who develop STF have a special MHC class constellation [33,34,47].

### 3.8. Comparison of CoDB and CoDA Graphs

When patients were classified according to their drawings of the CoD prior to BoNT therapy (CoDB graphs; RO, CO, and DO subgroups), no significant difference in distributions of the CoDA graphs across these three groups could be detected (Table 2). It is highly likely that the number of patients in the entire cohort was too small to detect significant differences. 

Only one patient (1/66 = 1.5%) drew a CoDB and a CoDA graph, which could not be classified according to the main curvature since he had experienced long-lasting periods of severe worsening, followed by relapses. This occurred before and after BoNT therapy. (See Figure 3, left and right). Obviously, this is a rare and special variant of CD. Longer-lasting relapses have been previously described in about 5% of the patients [48].

### 3.9. Recommendations Based on Differences in Response Behaviour

Treating physicians of BoNT-naïve CD patients should ask for the course of the severity of CD before BoNT-A therapy is initiated. Initial high doses should be avoided, but CD patients with a low initial severity should be treated with as high an initial dose as all other CD patients. Since no NAb induction under continuous therapy with the complex-protein-free BoNT-A preparation Xeomin^®^ (incoBoNT-A) has been reported so far [26], incoBoNT-A appears to have a lower antigenicity, as compared to the complex-protein-containing Botox^®^ or Dysport^®^ preparations. Consequently, BoNT-A therapy should be initiated with incoBoNT-A as the BoNT-A preparation with the lowest antigenicity [49].

Since NAb titres decrease under incoBoNT-A therapy [45], switching to Xeomin^®^ is recommended as soon as STF is suspected. In patients with STF, DBS operation should be postponed until several cycles of treatment with another BoNT-A preparation, especially the complex-protein-free incoBoNT-A preparation, have been tried.

### 3.10. Strengths and Limitations of the Study

The strength of the present pilot study is that, for the first time, patients򲀙 drawings of the course of CD severity after onset of BoNT therapy were analysed. They visualize the broad spectrum of response to BoNT and allow researchers to easily detect “golden responders” on the one hand, and on the other, “difficult to treat” patient subgroups, such as the primary nonresponders in the PR subgroup and the secondary nonresponders in the STF subgroup. 

However, the sample size of the present study was too small to allow for generalization and the characterization of special response-type subgroups of patients with a special course of disease before and after BoNT therapy. Therefore, further studies using patients’ drawings with more patients included are recommended.

## 4. Conclusions

This pilot study on CD patients´ drawings of their course of disease severity after the onset of BoNT therapy (CoDA graphs) demonstrates that this method can be used in clinical practise and provides helpful information on patients´ assessments of the efficacy of BoNT therapy and their response behaviours. Patients with primary and secondary treatment failure can easily be distinguished from well-responding patients by means of CoDA graph drawings. In addition, the response behaviour after switching to another BoNT-A preparation in patients with STF is clearly displayed. Furthermore, analysis of the treatment-related data of patients with different CoDA graphs indicates risk factors for the induction of NAbs. However, more frequent applications of this method are necessary to explore its usefulness in clinical practise.

## 5. Materials and Methods

The present study adhered to the guidelines for good clinical practice (GCP) and the Declaration of Helsinki and received approval from the local ethics committee of of the University of Düsseldorf (number: 4085).

### 5.1. Patients: Demographic and Treatment-Related Data

The present study is a combination of a cross-sectional study (assessment and treatment at the day of recruitment and a retrospective study heavily relying on patients’ recall of the course of the severity of their disease. This is demonstrated in a scheme presented in Figure 4.

The present study includes individuals who met the following criteria: (i) an age above 17; (ii) a diagnosis of idiopathic CD; (iii) provision of written informed consent; (iv) initiation of therapy at the Outpatient Department of the University of Düsseldorf (Germany) and reception of continuous treatment every 12 to 13 weeks without interruption of BoNT therapy for more than one treatment cycle; and (v) reception of at least three injections of BoNT. Exclusion criteria were as follow: (i) patients under legal care; (ii) patients with a multifocal or segmental dystonia; (iii) individuals with an additional disabling disease other than CD; and (iv) patients with a clinically overt disturbance of mood or perception.

The study screened over 300 charts of CD patients who were regularly treated with BoNT therapy. Patients who had interruptions of less than two treatment cycles were informed about the study while waiting in the outpatient department for their next injection. Out of these, 74 patients were consecutively recruited after they provided informed written consent and had met all inclusion and exclusion criteria.

The following demographic data were extracted from the charts of the patients: age at day of recruitment (AGE), age at onset of symptoms (AOS), age at onset of therapy (AOT), and duration of therapy (DURT). We also extracted the following treatment-related data: TSUI score at onset of therapy (ITSUI), initial BoNT preparation, and initial total dose (IDOSE). The time to therapy (DURS = time span during which patients had tolerated symptoms without resorting to BoNT therapy) was calculated as DURS = AOT − AOS. 

Patients were asked to rate the change in CD since the onset of BoNT-therapy as the percent of the severity of CD at the onset of therapy (IMPQ: improvement (positive value), worsening (negative value)). The treating physician scored the actual severity of the CD on the day of recruitment by means of the TSUI score [10] (ATSUI) and documented the BoNT-A preparation used, as well as the actual total dose (ADOSE). Improvement in the severity of CD was determined by means of the TSUI score (IMPTSUI), calculated as (ITSUI − ATSUI) × 100/ITSUI) and the increase of dose (INDOSE) during treatment as ADOSE-IDOSE. For the sake of comparison, doses of different preparations were transformed into unified doses by leaving doses of ona- and incobotulinum toxin unchanged, and by dividing abobotulinum toxin doses by 3, following a European consensus paper [50].

### 5.2. Drawing of the Course of Disease Graphs

The drawing of the course of disease graphs (CoD graphs) has been described in detail previously [9]. Patients were comfortably seated in front of a desk, one hand held a piece of paper with a square 10 cm × 10 cm in size, and the other hand held a pen. To draw the CoD graph before BoNT therapy (the CoDB graph), patients were instructed to draw a continuous graph representing their CoD severity of CD from the onset of symptoms until the onset of BoNT-A therapy by starting at 0 in the left lower corner (=onset of symptoms) and ending at the right upper corner (=100% = severity of CD at onset of BoNT-A therapy). For the drawing of the CoD graph after the onset of BoNT therapy (CoDA graph), patients were instructed to draw a continuous graph into a second square of the same size, representing their CoD severity of CD from the onset of BoNT-A therapy until the actual day of recruitment. As a first step, patients had to mark the actual severity of CD on the day of recruitment (ASCD) on a line through the right edge of the square. Then, a continuous graph had to be drawn from the left upper corner (=100% = severity of CD at the onset of BoNT-A therapy) to the ASCD mark on the right edge of the square. Three attempts for the CoDB and CoDA graph drawing were allowed, as well as verbal help provided by the investigator, but no drawing assistance was provided. No example of a CoD graph was shown so as to avoid any bias. 

The improvement of CD as estimated by the drawing of the CoDA graph (IMPD) was calculated as (10 − ASCD) × 10. 

After drawing the CoD graphs, patients underwent a detailed clinical investigation and were injected. 

### 5.3. Classification of the CoD Graphs

Out of 74 patients, 3 were unable to create a continuous CoDB and CoDA graph, while 5 others were unable to mark the ASCD or create a continuous CoDA graph, even after three attempts. Although their demographic and treatment-related data were included in the study (column “ALL” in Table 1), their graphs were excluded from further analysis.

A total of 71 CoDB and 66 CoDA graphs were scanned using a standard scanner. Commercially available software, Digitizeit^®^ [51], was used to digitize the graphs. The origin and end of the x-axis, the origin and end of the y-axis, and the origin and end of the CoD graphs had to be marked on the scan. The software produced an x–y table for each graph when a stick was moved along the scan from the origin to the end of the graph. These data were used to produce the digitized versions of the graphs, which were used for further analysis. 

The classification of CoDB graphs into three different types, depending on the main curvature, has been described previously [9]. When more than 75% of the CoDB graph was drawn above the 45° line from the lower left to the upper right corner, the graph was classified as “rapid onset” (RO)-type. When the graph oscillated around the 45° line, the graph was classified as “continuous onset” (CO)-type. When 75% of the graph was drawn below the 45° line, the graph was classified as “delayed onset” (DO)-type. Two CoDB graphs did not fit into these three categories and were classified as “other onset” (OO)-type. (For details, see [9]). 

The classification of the 66 digitized CoDA graphs was performed in a similar manner. In the first step, a line between the upper left corner and the ASCD mark was drawn. A graph which was drawn with more than 75% below this line was classified as “rapid response” (RR)-type; a graph which closely followed this line was classified as “continuous response” (CR)-type. A graph in 75% of the drawing remained above this line should have been classified as a “delayed response”-type to preserve consistency. However, since all of the graphs of this type had high ASCDs, this type was called “poor response” (PR)-type. Those RR-type graphs which had a clear U-shape with a secondary worsening after an initial good response were classified as “secondary treatment failure” (STF)-type. Those STF-type graphs with a secondary improvement after a secondary worsening were classified as STFII graphs, and the remaining STF graphs were classified as STFI graphs. Examples of these five types of CoDA graphs (RR, CR, PR, STFI, and STFII) are presented in Figure 5. One graph did not fit into these four categories and was classified as “other response” (OR)-type (not displayed in Figure 5).

CoDA graphs were classified by each of the authors (IS, MS, BÜ, and RB). In 10 discordant cases, final classification was achieved during a consensus meeting. 

A total of 10 of the 17 patients with an STF-type were included in another study on antibody formation in CD patients treated on a long-term basis. 

### 5.4. Statistics

The patients were divided into five subgroups based on the type of CoDA graphs they produced (RR, CR, PR, STF, and OR subgroups). ANOVA was conducted on AGE, AOS, DURS, DURT, IDOSE, ADOSE, INDOSE, ITSUI, ATSUI, IMPTSUI, IMPQ and IMPD to determine whether there were any significant differences among the RR, CR, PR, and STF subgroups (see Table 1). The OR subgroup, which consisted of only 1 patient, was excluded from the ANOVA tests. A chi^2^ test was used to determine if there were any differences in the sex distribution and BoNT-preparation switchers across patient subgroups.

The patients were also divided into three subgroups based on type of CoDB graph (RO, CO, and DO subgroups), and the distribution of CoDA types for each subgroup is presented in Table 2. A Friedman test was conducted to investigate whether there was a relationship between the CoDB and CoDA graphs. All statistical analyses were performed using the SPSS^®^ statistical package (version 25; IBM, Armonk, NY, USA).

## Figures and Tables

**Figure 1 toxins-15-00431-f001:**
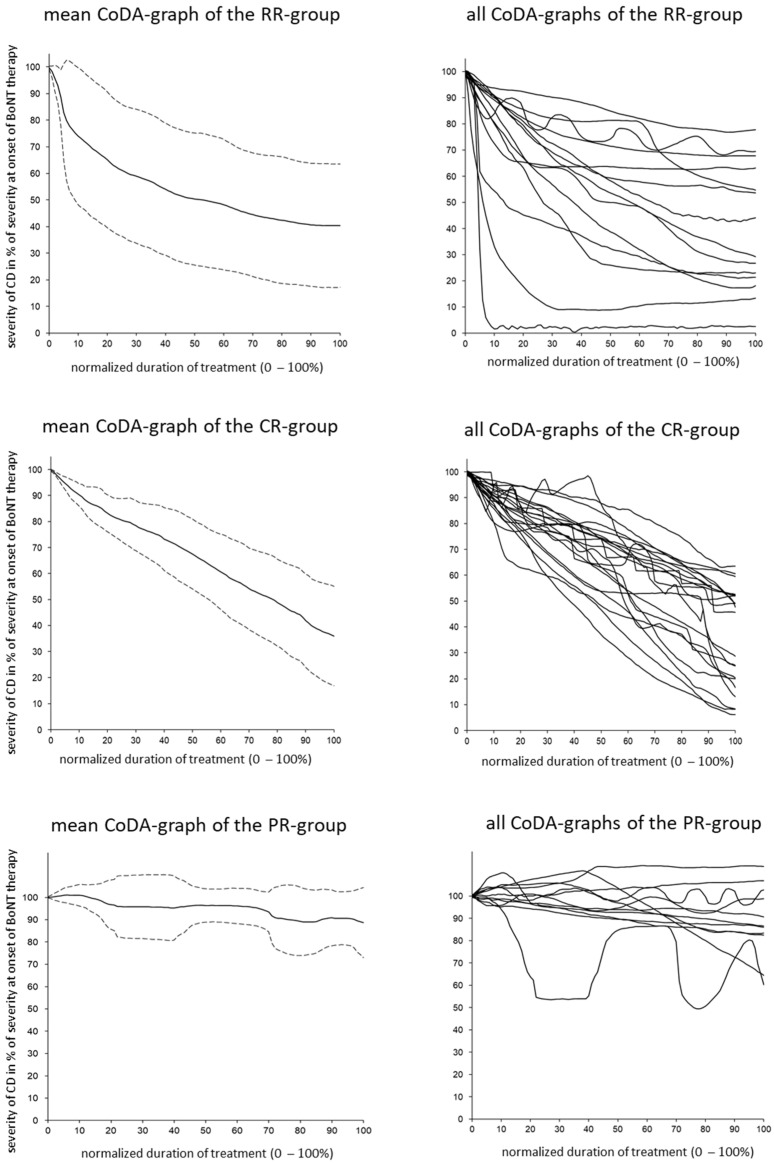
(**Left side**) Average CoDA graphs and the corresponding plus/minus 1 standard deviation ranges across all patients within the rapid response (RR), continuous response (CR), and poor response (PR) subgroups are presented. (**Right side**) All CoDA graphs of all patients within the RR, CR, and PR subgroups are presented.

**Figure 2 toxins-15-00431-f002:**
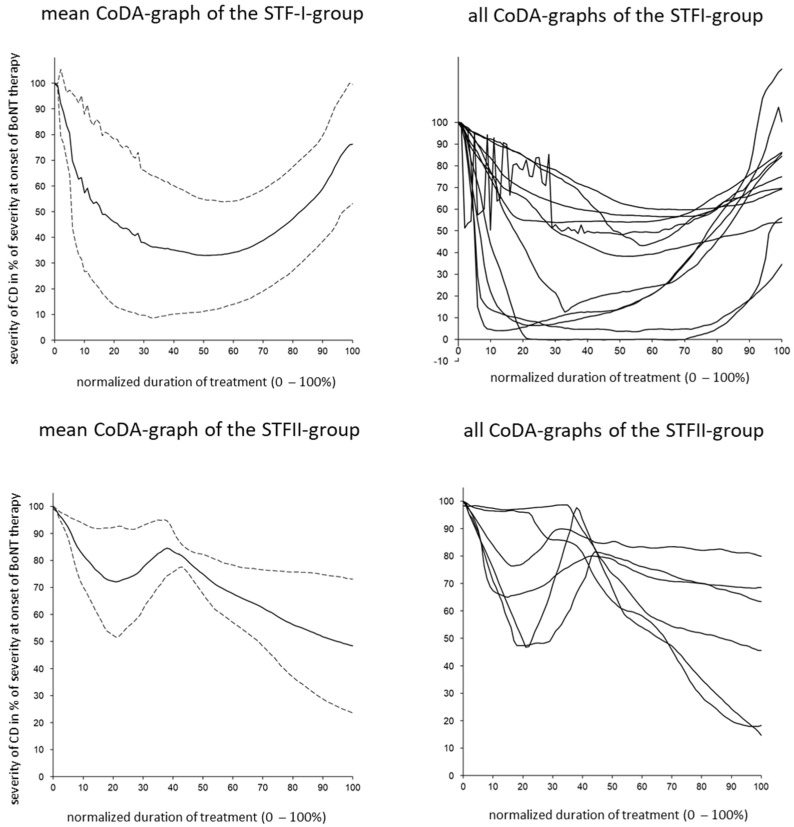
(**Left side**) Average CoDA graphs and the corresponding plus/minus 1 standard deviation ranges across all patients within the secondary treatment failure I (STFI) and the secondary treatment failure II (STFII) subgroups are presented. (**Right side**) All CoDA graphs of all patients within the STFI and STFII subgroups are presented.

**Figure 3 toxins-15-00431-f003:**
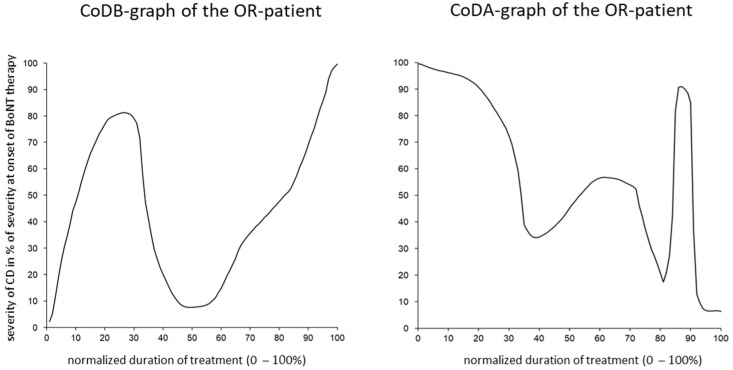
The CoDB graph (**left side**) and the CoDA graph (**right side**) of one exceptional patient in the other response (OR) subgroup are presented. Before BoNT therapy, as well as after BoNT therapy, this patient had experienced long-lasting relapses, which is a rare phenomenon in CD.

**Figure 4 toxins-15-00431-f004:**
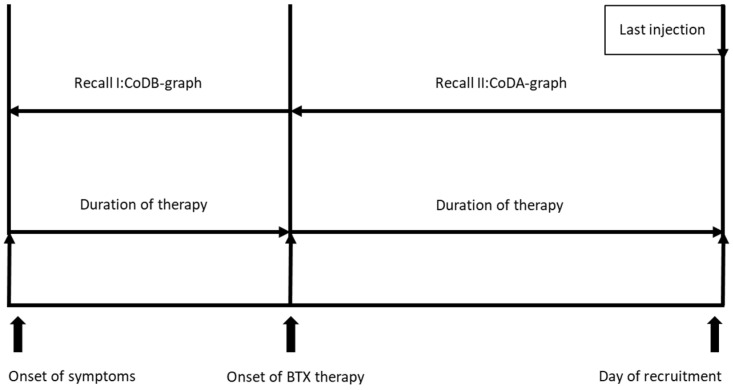
Treatment-related data were produced on the day of recruitment and were extracted from the charts of the patients. Drawings of the CoDB and CoDA graphs are based on patients’ recall on the day of recruitment.

**Figure 5 toxins-15-00431-f005:**
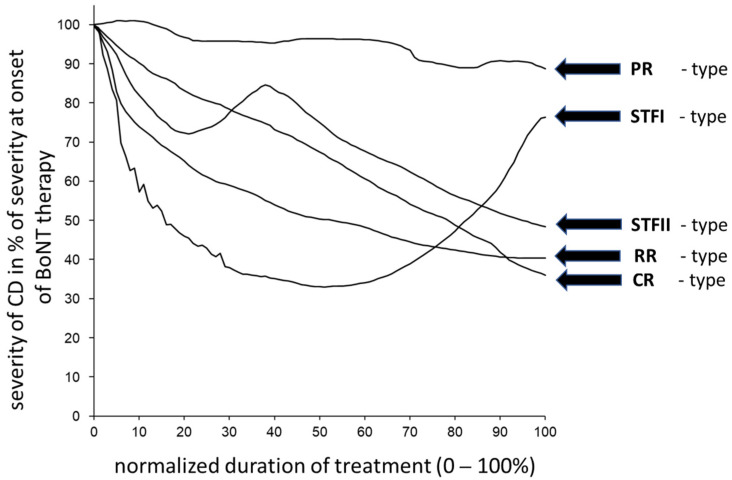
Five types of CoDA graphs (RR-, CR-, PR-, STFI-, and STFII-type).

**Table 1 toxins-15-00431-t001:** Demographic and treatment-related data and outcome measures in the four subgroups (RR, CR, PR, and STF) and the entire cohort (ALL).

Parameter		RR	CR	PR	STF	ALL	Significance-Level *p* < 0.05
n		15	21	12	17	74	
female/male		13/2	14/7	8/4	8/9	49/25	0.13; n.s.
AGE (years)	MV/SD	64.0/11.4	61.7/12.1	59.1/11.6	57.8/9.8	60.2/11.6	0.43; n.s.
	MIN–MAX	28.7–76.5	41.9–87.0	41.7–81.3	43.4–73.8	28.7–87.0	
AOS (years)	MV/SD	46.9/13.9	46.9/10.8	45.0/10.7	43.0/15.0	45,26/12.5	0.78; n.s.
	MIN–MAX	20.8–66.3	25.9–64.6	34.0–73.4	14.7–60.3	14.7–73.4	
DURS (months)	MV/SD	78.8/125.0	48.5/66.7	93.1/106.2	59.6/77.9	68.9/91.5	0.55; n.s.
	MIN–MAX	2.9–438.2	2.0–254.5	1.0–324.2	5.0–281.2	1.0–438.2	
DURT (months)	MV/SD	126.0/80.1	129.8/77.1	75.3/74.9	118.0/81.8	115.7/80.3	0.26; n.s.
	MIN–MAX	16.0–274.2	15.0–270.3	0.6–282.3	24.0–321.2	0.6–321.2	
IDOSE (uDU)	MV/SD	151.4/47.7	161.8/58.3	172.0/124.1	184.8/102.4	166.3/80.6	0.73; n.s.
	MIN–MAX	100.0–250.0	31.3–300.0	75.0–500.0	31.3–450	31.3–500.0	
ADOSE (uDU)	MV/SD	167.3/86.0	233.0/98.4	227.1/132.2	249.4/125.9	217.9/114.7	0.21; n.s.
	MIN–MAX	50.0–400.0	50.0–400.0	100.0–500.0	15.0–450.0	15.0–500.0	
INDOSE (uDU)	MV/SD	13.4/85.1	65.1/74.9	47.8/73.5	71.8/91.1	50.6/87.2	0.24; n.s.
	MIN–MAX	−110.0–275.0	−15.0–275.0	−15.0–200.0	−15.0–275	−110.0–275.0	
ITSUI	MV/SD	11.0/2.0	8.5/2.3	6.0/2.0	9.8/1.3	8.9/2.4	0.10; n.s.
	MIN–MAX	8.0–13.0	5.0–12.0	4.5–8.5	8.0–11.0	4.0–13.0	
ATSUI	MV/SD	3.1/2.7	4.0/3.1	4.6/1.8	5.5/2.1	4.4/2.6	0.07; n.s.
	MIN–MAX	0.0–8.0	0.0–10.0	2.0–8.0	2.0–9.0	0.0–10.0	
IMPTSUI	MV/SD	7.9/2.5	4.7/3.9	1.6/2.0	3.9/1.8	3.9/3.8	0.17; n.s.
	MIN–MAX	2.0–13.0	0.0–11.0	−2.0–5.0	−2.0–8.0	−2.0–13.0	
IMPQ	MV/SD	58.7/25.3	60.7/17.9	11.3/19.6	32.7/29.9	42.9/31.3	*p* < 0.009
	MIN–MAX	10.0–90.0	40.0–90.0	−20.0–50.0	−30.0–80.0	−30.0–90.0	
IMPD	MV/SD	61.3/24.0	64.0/19.6	7.0/20.3	34.8/30.1	46.0/32.0	*p* < 0.009
	MIN–MAX	21.7–98.0	38.7–92.3	−30.0–40.9	−25.8–84.8	−30.0–98.0	

Note: RR = rapid response subgroup; CR = continuous response subgroup; PR = poor response subgroup, STF = secondary treatment failure subgroup; ALL = data from all recruited patients (for details, see the Section 5); MV = mean value; SD = standard deviation; uDU = unified dose units AGE = age at recruitment; AOS = age at onset of symptoms; DURS = time from onset of symptoms to BoNT therapy; DURT = duration of BoNT therapy; IDOSE = dose at onset of BoNT therapy; ADOSE = dose at investigation (actual dose); INDOSE = increase of dose during BoNT therapy; ITSUI = TSUI score at onset of BoNT therapy; ATSUI = actual TSUI at day of recruitment; IMPTSUI = improvement according to TSUI score; IMPQ = improvement of symptoms according to questionnaire; IMPD = improvement according to drawing (for details, see the Section 5). n.s.=not significant.

**Table 2 toxins-15-00431-t002:** Number of different types of CoDB graphs across the different CoDA graph groups.

	CoDB Graphs						
CoDA Graphs	RO	CO	DO	OO	Good CoDB Graph	No CoDB Graph	All Patients
RR	2	8	10	1	21		21
CR	6	3	6		15		15
PR	1	4	7		12		12
STF	5	6	6		17		17
OR				1	1		1
Good CoDA graph	14	21	29	2	66		
No CoDA graph	2	2	1			3	8
All patients	16	23	30	2		3	74

Note: RO = rapid onset subgroup; CO = continuous onset subgroup; DO = delayed onset subgroup; RR = rapid response subgroup; CR = continuous response subgroup; PR = poor response subgroup; STF = secondary treatment failure subgroup.

## Data Availability

Data are only available upon request due to restrictions of privacy or ethics. The data presented in this study are available on request from the corresponding author.

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
