# Peer review of "Lessons about Botulinum Toxin A Therapy from Cervical Dystonia Patients Drawing the Course of Disease: A Pilot Study"

_toxins, 2023, doi:10.3390/toxins15070431_

Round 1

Reviewer 1 Report

The article aimed to compare the course of severity of cervical dystonia before and after long-term botulinum toxin therapy. It involved 74 outpatients with idiopathic cervical dystonia who received continuous botulinum toxin treatment and had received at least three injections. The patients were asked to draw the course of cervical dystonia severity from symptom onset until botulinum toxin therapy (CoDB-graph) and from botulinum toxin therapy initiation until recruitment (CoDA-graph). The researchers analyzed the different types of graphs and extracted demographic and treatment-related data from patient charts.

The study found that patients who had a clear and rapid initial response to botulinum toxin therapy had the best outcomes. These patients received lower doses of botulinum toxin and had fewer switches between different botulinum toxin preparations. On the other hand, patients who initially improved but later experienced a secondary worsening had the worst outcomes. These non-responders received higher initial and ongoing doses of botulinum toxin and had frequent switches between different preparations. Additionally, there were patients who did not experience any improvement at all.

The study concluded that patients' drawings of the course of disease severity could help identify primary and secondary non-responders to botulinum toxin therapy, as well as "golden responders" who showed a positive response. The use of high initial doses was confirmed as a risk factor for the development of secondary non-response.

Sample Size and Selection: The study included a limited number of outpatients (74) with idiopathic cervical dystonia who were continuously treated with botulinum toxin. The sample size may limit the generalizability of the findings, and the criteria used for patient selection may introduce bias.

Retrospective Design: The study appears to have a retrospective design since it involved extracting data from patient charts. Retrospective studies have inherent limitations, such as potential inaccuracies in data collection and reliance on existing records.

Lack of Control Group: The study does not mention the inclusion of a control group for comparison. Without a control group, it may be challenging to determine the true effects of long-term botulinum toxin therapy and establish causal relationships.

Reliance on Patient-Reported Drawings: The study relied on patients' drawings to assess the course of cervical dystonia severity. The subjective nature of this assessment method may introduce variability and subjectivity into the results.

Lack of Detailed Treatment Information: While the study mentions extracting treatment-related data from patient charts, it does not provide specific details about the botulinum neurotoxin treatment protocols used, including injection sites, dosages, or injection frequencies. These details are important for understanding the treatment outcomes and identifying potential confounding factors.

What are the injection points and doses taken, anatomical information should have given for toxin injections. Please regard and cite “Effective botulinum toxin injection guide for treatment of cervical dystonia” and “Anatomical guide for botulinum neurotoxin injection: Application to cosmetic shoulder contouring, pain syndromes, and cervical dystonia

Lack of Follow-up Period: The abstract does not mention the duration of the follow-up period after the onset of BoNT therapy. The absence of this information makes it difficult to evaluate the long-term outcomes of the treatment and assess its sustainability.

Overall, well written and I would like to give major revision. Should state these limitation in the end of the discussion section.

Minor errors.

Author Response

The article aimed to compare the course of severity of cervical dystonia before and after long-term botulinum toxin therapy. It involved 74 outpatients with idiopathic cervical dystonia who received continuous botulinum toxin treatment and had received at least three injections. The patients were asked to draw the course of cervical dystonia severity from symptom onset until botulinum toxin therapy (CoDB-graph) and from botulinum toxin therapy initiation until recruitment (CoDA-graph). The researchers analyzed the different types of graphs and extracted demographic and treatment-related data from patient charts.

The study found that patients who had a clear and rapid initial response to botulinum toxin therapy had the best outcomes. These patients received lower doses of botulinum toxin and had fewer switches between different botulinum toxin preparations. On the other hand, patients who initially improved but later experienced a secondary worsening had the worst outcomes. These non-responders received higher initial and ongoing doses of botulinum toxin and had frequent switches between different preparations. Additionally, there were patients who did not experience any improvement at all.

The study concluded that patients' drawings of the course of disease severity could help identify primary and secondary non-responders to botulinum toxin therapy, as well as "golden responders" who showed a positive response. The use of high initial doses was confirmed as a risk factor for the development of secondary non-response.

Sample Size and Selection: The study included a limited number of outpatients (74) with idiopathic cervical dystonia who were continuously treated with botulinum toxin. The sample size may limit the generalizability of the findings, and the criteria used for patient selection may introduce bias.

Retrospective Design: The study appears to have a retrospective design since it involved extracting data from patient charts. Retrospective studies have inherent limitations, such as potential inaccuracies in data collection and reliance on existing records.

Lack of Control Group: The study does not mention the inclusion of a control group for comparison. Without a control group, it may be challenging to determine the true effects of long-term botulinum toxin therapy and establish causal relationships.

Reliance on Patient-Reported Drawings: The study relied on patients' drawings to assess the course of cervical dystonia severity. The subjective nature of this assessment method may introduce variability and subjectivity into the results.

Lack of Detailed Treatment Information: While the study mentions extracting treatment-related data from patient charts, it does not provide specific details about the botulinum neurotoxin treatment protocols used, including injection sites, dosages, or injection frequencies. These details are important for understanding the treatment outcomes and identifying potential confounding factors.

What are the injection points and doses taken, anatomical information should have given for toxin injections. Please regard and cite “Effective botulinum toxin injection guide for treatment of cervical dystonia” and “Anatomical guide for botulinum neurotoxin injection: Application to cosmetic shoulder contouring, pain syndromes, and cervical dystonia

Lack of Follow-up Period: The abstract does not mention the duration of the follow-up period after the onset of BoNT therapy. The absence of this information makes it difficult to evaluate the long-term outcomes of the treatment and assess its sustainability.

Overall, well written and I would like to give major revision. Should state these limitation in the end of the discussion section.

The authors are thankful to all three reviewers for their helpful comments and suggestions.

We have addressed all points raised by the reviewers.

The first three sections of the comments of reviewer 1 nicely summarize our manuscript and its message.

These are two relevant points:

Reviewer 1 is right: The number of 74 patients is rather small. Differences between subgroups should therefore not be generalized until they are supported by a larger study. This is mentioned now.

In the present cohort patients were recruited consecutively. The selection criterion that only patients were recruited without interruptions of treatment of more than 1 treatment cycle avoids bias from incompliant patients.

This study heavily relies on patients recall of their course of disease from onset of symptoms until day of recruitment (=end of therapy since patients received their last injection at the day of recruitment). This is made clear by means of a graphic abstract.

A control group had to include CD-patients who did not receive BoNT treatment. These patients are either only mildly affected or undergo DBS-operation or have other reasons which make it difficult to compare such patients to BoNT treated patients. In our institution such patients are extremely rare so that it is impossible to recruit a control group.

 We are aware that patients´ data may be subjective and variable. Nevertheless, the CoD-drawings had not been random curves, but could be classified. Furthermore, we compared patients´ data with treatment related data produced by the treating physicians.

Reviewer 1 is right that information on injection sites, dosages and injection frequencies are important for the understanding of outcome. Therefore, we have presented some of these data in the tables. Furthermore, our team has experience in the treatment of CD-patients for more than 35 years, has developed the CAP/COL-concept and is familiar with all guidance techniques. The presentation of the type of CD, injection protocols, muscle patterns involved and dose per muscle would by far exceed the aim of the present study.

We now regard and cite these two mentioned papers, which are very helpful for therapy optimization.    

The mean duration of treatment is now not only mentioned in the tables and results but also in the abstract. The mean duration of follow-up and its variability (standard deviation) is also presented in detail for all subgroups in the tables.  

We have taken into account all comments and suggestions of the reviewers and have substantially revised the manuscript.

Reviewer 2 Report

Title:

Please make the title more concise, it is quite long at present.

Abstract:

Why did the Patients draw the severity  from onset of BoNT-therapy until day of recruitment (CoDA-graph) instead of until the end of therapy? 

Introduction:

In lines 57-59, please mention other types of treatments used in CD for patients not receiving BoNT/A.

In lines 67-68, please could the authors provide the specific BoNT dosing guidances based on standardized scales using the TSUI score or the Toronto Western Spasmodic Torticollis (TWSTRS)to adjust the choice of preparation, dosing regimen, and injection scheme?

Line 91: the dose of 500UI of BoNT/A is very high, please provide the usual dose range used for CD.

Line 105: please explain why the patients were not assessed for the full duration of the study but only until the recruitment into the study.

Results:

Section 2.2: please formulate results description with words rather than only abbreviations.

General comment: please provide the sub-groups definitions in the Results section to help the reader.

Also: please provide the calculation of normalized duration of treatment. It would be interesting to draw graphs with actual months of treatment. (Month 0 to Month 12)

Lines 135-137: Please repeat the abbreviations definitions in the legend of the Fig.1;

Lines 139-141: The same for Fig 2: please provide the definition of the abbreviations.

Lines 155, 159 and 165: please specify if the patients were switched to BoNT/B and the reasons for switching. Did those patients show any immunity against BoNT/A?

Lines 182-183: Please provide more details on the hemidiaphragm test method and data. Please write "hemidiaphragm" instead of "hemidiaphragma".

Line 198-199: Please provide one sentence at least to explain the choice of the "stair case model".

Lines 209 and 223: Please elaborate more about the different factors influencing the patients response rate to BoNT treatments.

Line 260-261: Please provide the differences in the BoNT formulations (that may partly explain the different responses).

Line 287: please remove space between "v" and "e" in the word "severe".

Line 291: "Treating physicians of BoNT naïve CD-patients should ask" replace with : "Physicians initiating the treatment of BoNT naïve CD-patients should..."

Line 312: recommended to assess the patients for the full BoNT therapy duration in further studies.

Conclusion:

It is important also to include additional treatments possibly taken by the patients to treat pain or CD in general. In the discussion, it would be very useful for the clinician to give more background on the studies describing treatment failure subsequent to antibodies responses. 

The manuscript is very well written, has an important clinical message, and should be of great interest to the readers and the clinicians.

Author Response

Title:

Please make the title more concise, it is quite long at present.

Abstract:

Why did the Patients draw the severity  from onset of BoNT-therapy until day of recruitment (CoDA-graph) instead of until the end of therapy? 

Introduction:

In lines 57-59, please mention other types of treatments used in CD for patients not receiving BoNT/A.

In lines 67-68, please could the authors provide the specific BoNT dosing guidances based on standardized scales using the TSUI score or the Toronto Western Spasmodic Torticollis (TWSTRS)to adjust the choice of preparation, dosing regimen, and injection scheme?

Line 91: the dose of 500UI of BoNT/A is very high, please provide the usual dose range used for CD.

Line 105: please explain why the patients were not assessed for the full duration of the study but only until the recruitment into the study.

Results:

Section 2.2: please formulate results description with words rather than only abbreviations.

General comment: please provide the sub-groups definitions in the Results section to help the reader.

Also: please provide the calculation of normalized duration of treatment. It would be interesting to draw graphs with actual months of treatment. (Month 0 to Month 12)

Lines 135-137: Please repeat the abbreviations definitions in the legend of the Fig.1;

Lines 139-141: The same for Fig 2: please provide the definition of the abbreviations.

Lines 155, 159 and 165: please specify if the patients were switched to BoNT/B and the reasons for switching. Did those patients show any immunity against BoNT/A?

Lines 182-183: Please provide more details on the hemidiaphragm test method and data. Please write "hemidiaphragm" instead of "hemidiaphragma".

Line 198-199: Please provide one sentence at least to explain the choice of the "stair case model".

Lines 209 and 223: Please elaborate more about the different factors influencing the patients response rate to BoNT treatments.

Line 260-261: Please provide the differences in the BoNT formulations (that may partly explain the different responses).

Line 287: please remove space between "v" and "e" in the word "severe".

Line 291: "Treating physicians of BoNT naïve CD-patients should ask" replace with : "Physicians initiating the treatment of BoNT naïve CD-patients should..."

Line 312: recommended to assess the patients for the full BoNT therapy duration in further studies.

Conclusion:

It is important also to include additional treatments possibly taken by the patients to treat pain or CD in general. In the discussion, it would be very useful for the clinician to give more background on the studies describing treatment failure subsequent to antibodies responses. 

The manuscript is very well written, has an important clinical message, and should be of great interest to the readers and the clinicians.

Here is our attempt to modify the title. We have shortened the original title, but reviewer 3 suggested a further extension. Reviewer 2 and the editor may decide which is the best version of the title.

At the day of recruitment patients received their last injection. Thus, the day of recruitment is the end of therapy. This is made more clearly in the graphic abstract (Fig. 4).

 Muscle relaxing and pain medication and physiotherapy are mentioned now.

Optimization of therapy is a challenging task and handled differently in different centers. But this is an important point and is now addressed in the introduction.

500 U aboBoNT/A is a low standard dose of Dysport® (see reference [19]).

We apologize for this confusion. This point is made clear now by means of a graphic abstract (Fig. 4).

Reviewer 2 is right: the abbreviations of the subgroups have not been explained previously. This is corrected now.

Now corrected!

All durations of treatment were set to 10 (Time=t*10/DURT). We have performed plots with duration of treatment as x-axis. Because of the large variation of DURT CoDA-graphs could not be compared anymore. Therefore, duration of treatment had to be normalized.

 Preparation

This is improved.

This is improved.

This is an important point and is addressed now in detail. In our center, a patient is switched to another BoNT/A when (i) he had developed a secondary treatment failure (ii) because of private reasons, and (iii) because of cost aspects. In Germany it is possible to buy a BoNT/A-preparation for a much cheaper price in another near-by country.

NABs were tested in a cross-sectional study, but usually not in individual patient during routine clinical practice.

This point is addressed.

Patients experience an improvement injection by injection (step by step). The severity declines step by step. Therefore, we talk about “stair case”. But this is not a special model.

This point is addressed now.

This point is addressed now.

We are thankful for this careful observation.

We are also thankful for this improvement.

This point is clarified now (Fig. 4).

It has been reported repeatedly that only in 50% of the patients with suspected STF neutralizing antibodies can be detected. In the present study we have not analyzed NABs systematically, we therefore focus on the clinical aspect of STF.  Papers on NAB induced STF are mentioned in the references [33,34,35,37,38].

Reviewer 3 Report

An interesting article, well-designed, well-conducted, and well-written. Strengths and limitations of the study have been acknowledged adequately.  I suggest to include in the title "a pilot study" as a minor correction

Author Response

An interesting article, well-designed, well-conducted, and well-written. Strengths and limitations of the study have been acknowledged adequately.  I suggest to include in the title "a pilot study" as a minor correction

The authors are thankful for these positive comments. “A pilot study” is now included in the title.

Round 2

Reviewer 1 Report

The authors have well described answered the questions.